

# A 7-year record of vertical profiles of radar measurements and precipitation estimates at Dumont d'Urville, Adélie Land, East Antarctica

Valentin Wiener[1], Marie-Laure Roussel[1], Christophe Genthon[1], Étienne Vignon[1], Jacopo Grazioli[2], and Alexis Berne[2]

[1]Laboratoire de Météorologie Dynamique, Institut Pierre-Simon Laplace, Sorbonne Université/CNRS/École Polytechnique - IPP, Paris, France
[2]Environmental Remote Sensing Laboratory, Swiss Federal Institute of Technology in Lausanne, Lausanne, Switzerland

**Correspondence:** Valentin Wiener (valentin.wiener@lmd.ipsl.fr)

**Abstract.** Solid precipitation measurements in Antarctica are crucial as snowfall represents the main water input term for the polar cap, and its probable increase in the coming century can mitigate sea-level rise caused by global warming. This paper presents 7 years of Micro Rain Radar (Metek MRR-2) data at the Dumont d'Urville station in coastal Adélie Land, East Antarctica. Statistics are calculated on 3 radar variables (equivalent reflectivity, mean Doppler velocity and signal-to-noise ratio) to outline the main characteristics of the radar dataset. Seasonal and interannual variabilities are also investigated, but no significant temporal trends are detected except for the seasonal mean Doppler velocity which is higher in summer and lower in winter.

We then use the snowfall rate ($S$) data from a colocated snow-gauge to estimate the MRR precipitation profile from the radar equivalent reflectivity ($Ze$) through a locally derived Ze-S relation. We find the relation $Ze = 43.3\ S^{0.88}$. The processing method used to obtain this relation, data quality and uncertainty considerations are discussed in the manuscript.

In order to give an example of application of the dataset, a brief statistical comparison of the MRR precipitation rate along the vertical with model data from the ERA5 reanalysis and the LMDZ climate model is performed, and notably shows that models underestimate heavy precipitation events.

## 1 Introduction

Precipitation is the largest positive term for the mass balance of the Antarctic ice sheet. It originates from evaporation over the surrounding oceans, advection of water vapor to and over the ice sheet by the atmospheric circulation, then condensation and fall of condensed water to the surface. Various atmospheric processes determine how much water vapor effectively condensates above the ice sheet, how much falls all the way down to the surface, and how much remains at the surface and effectively contributes to accumulation and thus the ice sheet mass balance. Thomas et al. (2017) estimated from ice cores that the Antarctic surface mass balance have increased in average by $14 \pm 2.8$ Gt per decade since 1900.



Even concentrating on precipitation proper (ignoring post-deposition processes) raises a number of issues that may be ignored in the most direct approaches to precipitation studies. In fact, many studies, and many available climatologies of precipitation, focus on precipitation at the surface (e.g. the Global Precipitation Climatology Project (GPCP), Adler et al. (2018)).This is sensible, as many issues with precipitation relate to the surface water budget, water resources at the surface and mass balance of continental water bodies such as lakes and ice caps. One practical reason for concentrating on surface precipitation is that it can be measured with ground-based instruments such as snow gauges (e.g., Seefeldt et al. (2021)). The fate of precipitation in the atmospheric column, on the other hand, is more elusive due to the difficulty of monitoring the atmosphere along the vertical dimension. Satellite-borne radars such as CloudSat, which operated from 2006 to 2011, enabled the observation of precipitation above Antarctica on a continental scale (Palerme et al. (2014)), but with limited temporal resolution (one orbit every 5 days), limited spatial coverage (north of 82°S), and without information below 1300 m above ground level. Grazioli et al. (2017b) showed the importance of low-level processes such as the sublimation of precipitation due to the dry air flowing from the Antarctic plateau, which was estimated to reduce snowfall by 17% in average all over the continent. This process can bias satellite estimates of precipitation at the surface, and raises the necessity of ground-based measurements of the atmospheric column.

Ground-based remote sensing using profiling techniques, such as meteorological radars and lidars, can provide valuable additional data. Instrumental challenges (cost, technical expertise, energy requirements) tend to limit these applications to specific sites and contexts, e.g. for operational meteorology, or weather and hydrological risk predictions. The recent availability of affordable, compact, low consumption, relatively easy to use precipitation profiling radars has been a game changer for the study of the Antarctic precipitation in the atmospheric column rather than at the surface only. This has opened the possibility of studying and documenting the processes occurring in the atmospheric column, from which surface precipitation results. This has important added value for understanding the precipitation physics and evaluating meteorological and climate models. Numerical models are essential tools to forecast future climate changes, including the future contribution of Antarctic precipitation evolution to global sea-level change. If Antarctica was to melt entirely, global sea-level would rise by more than 60 meters. This will not happen in a foreseeable future but the fact that realizing just 1% of this potential would raise global sea-level by 60 cm is a major source of concern (IPCC, Pörtner et al. (2022)). Conversely, an increase in accumulation due to increasing precipitation over the continent, which is for example estimated at $51 \pm 11$ Gt yr$^{-1}$ between 1991 and 2005 by Lenaerts et al. (2018), has a mitigating effect on sea-level rise. Medley and Thomas (2019) estimated that precipitation over Antarctica moderated sea-level rise by 2.5 mm per decade since 1979.

In recent years, Antarctica has significantly benefited from new approaches to observe precipitation. Measuring solid precipitation using traditional gauge methods is difficult (see SPICE project, Nitu et al. (2018)). It is particularly difficult in Antarctica where the main issues of solid precipitation measurements are exacerbated : strong winds in the peripheral regions (Turner et al. (2009) found more than 60 wind events of storm force or larger per year, i.e. over 24.5 m s$^{-1}$, in three coastal stations), very low precipitation rates in the interior (estimated at 36 mm yr$^{-1}$ above 2250 m and north of 82°S by Palerme et al. (2014)), problems with frost deposition on instruments, and low temperatures impacting electronic components.





Over the last decade, several research groups have deployed new generation of light precipitation radars at Antarctic stations: at Princess Elisabeth (longitude : 23.4, latitude : -72.0, altitude : 1392 m a.s.l., deployed in 2010, mostly seasonal, Gorodetskaya et al. (2015)), Mario Zucchelli (longitude : 164.1, latitude : -74.7, altitude : 10 m a.s.l., deployed in December 2016, mostly seasonal, Bracci et al. (2022)), Dumont d'Urville (140.0, -66.7, 41 m a.s.l., started in November 2015, Grazioli et al. (2017a),

Grazioli et al. (2017b), Genthon et al. (2018)), and other stations. Radars are not influenced by most of the problems that affect the measurement of solid precipitation with gauges but there are other issues. The main one is that indirect information, such as radar reflectivity resulting from the backscattering of microwaves by hydrometeors, has to be converted into hydrometeor distributions and concentrations in the atmosphere, then to mass and fall speed to retrieve a precipitation flux. This involves hypotheses and tuning. In addition, these radars have been initially developed, and are provided with processing tools, designed

for liquid precipitation. Obviously, for Antarctica, this has to be revised to access solid precipitation.

In this paper, we present 7 years of vertical profiling of precipitation at the Dumont d'Urville station in Adélie Land, East Antarctica using a Metek Micro-Rain Radar (MRR-2) precipitation profiler. The setting, instruments, data processing methods and datasets are presented in Sect. 2. The main characteristics of the MRR dataset, including variability, statistics and extremes

are presented in Sect. 3.1. In Sect. 3.2, the mean MRR snowfall profile is derived from an empirical and local Ze-S relation, enabling a vertical comparison with two climate models in section 3.3 as an example of application of the dataset. A general conclusion with information on data access and format is provided in Sect. 4.

## 2  Setup, data and methods

### 2.1  Micro Rain Radar

A Micro Rain Radar (Metek MRR-2, see Fig. 2a) transmitting in the K-band at 24 GHz was deployed at the Dumont d'Urville Antarctic station in late 2015. Grazioli et al. (2017a) and Genthon et al. (2018) describe the setting, processing and first set of data from the instrument. The Dumont d'Urville (DDU) station is located on the Petrels Island (140.0014, -66.6628, 41 m a.s.l., see Fig. 1), about 5 km off the coast of Adélie Land. As precipitation is essentially associated with synoptic extra-tropical cyclones there (Jullien et al. (2020)), observations at DDU are representative of precipitation at the nearby coast of the

Antarctic ice sheet. Setting the radar at DDU rather than on the ice sheet has the advantages that it provides easy permanent access to power and network, as well as servicing if necessary. It is installed within a radome which protects the instruments from the fierce winds that blow in the region. On the other hand, the radome induces some attenuation of the radar transmitted and reflected electromagnetic waves and thus some reduction of the sensitivity. This is in particular discussed in Grazioli et al. (2017a) that describes the first year of MRR data, Durán-Alarcón et al. (2019) that compares 2 years of the DDU MRR to

another MRR deployed at the Princess Elizabeth station (longitude : 23.35, latitude : -71.95), and lastly in Roussel et al. (2023) that presents an analysis of precipitation at DDU during the YOPP (Year Of Polar Prediction) southern hemisphere special observing period. A consequence of good operating conditions at DDU station is that, to the authors' knowledge and albeit not



the first MRR deployed in Antarctica, the DDU MRR offers the longest quasi continuous data series for the Antarctic region so far.

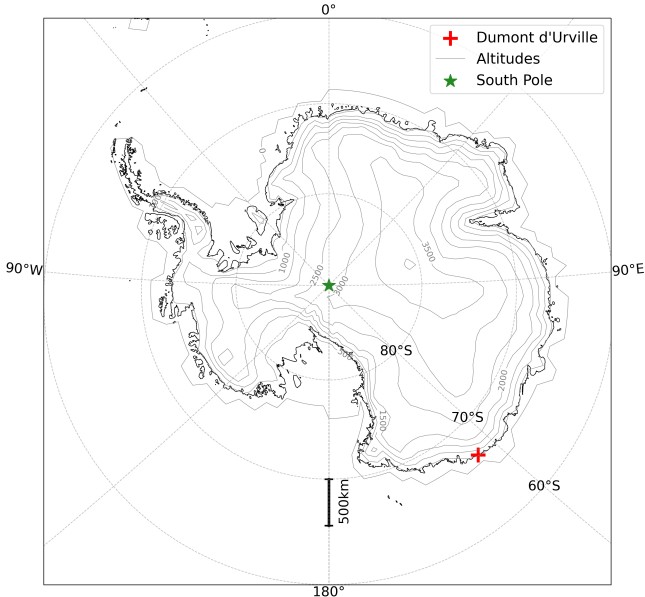

**Figure 1.** Topography of the Antarctic ice sheet and location of the Dumont d'Urville station.

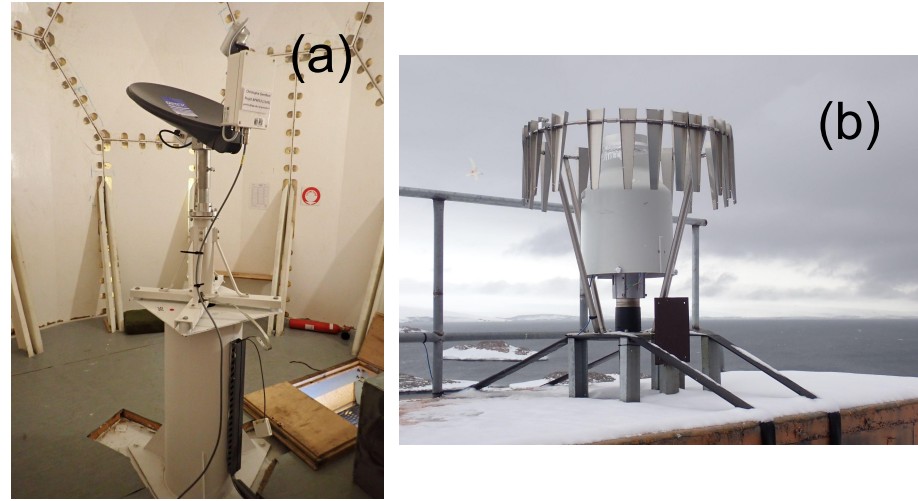

**Figure 2.** Picture of the DDU MRR in its radome (a) and the DDU snow-gauge (b).

As most of precipitation at DDU falls as snow, the MRR data have been processed using the Maahn and Kollias (2012) processing algorithm for snowfall, hereafter referred to as MK12. MK12 is especially suited for low signal-to-noise ratio (SNR)



measurements such as those obtained in snowfall, as it improves noise removal and allows the detection of weak updrafts thanks to a dynamic dealiasing procedure. Equivalent reflectivity Ze in dBz, mean Doppler velocity W in m s$^{-1}$ and SNR in dB are derived from the MRR minute-averaged raw spectrum, providing profiles up to 3 km a.g.l. with a vertical resolution of 100 m.

The two lowest and highest gates (below 300 m and above 2900 m) are considered too noisy by the algorithm and are therefore discarded from this study. A quality flag described in the metadata is also provided to give information about masked spectra not kept by the MK12 processing algorithm. We define the mean Doppler velocity as positive downward.

Due to various technical issues, the MRR was out of operation in about 6.4% of the total number of minutes in the record. This occurred e.g. during maintenance or power outages. Furthermore, 0.1% of the precipitating timesteps (i.e., when the

minute reflectivity is not null along the vertical) are discarded as the quality flag indicates a raw spectrum dealiasing failure. Lastly, equivalent reflectivities lower than -5 dBz are also discarded, in accordance with the threshold recommended by Maahn and Kollias (2012), removing 10.5% of precipitating 1-min timesteps. For a full description of this method, we refer the reader to Maahn and Kollias (2012).

The MRR being located inside a radome not optimized for the K-band, the signal is significantly attenuated. This was quan-

tified by Grazioli et al. (2017a) by comparison with a nearby X-band radar. The regression between the measured reflectivity values at X and K bands exhibited a slope close to 1, and an offset of about 6 dBz (to be added to the MRR data). This offset was confirmed by Durán-Alarcón et al. (2019) by comparison with a second MRR deployed at DDU outside of the radome for a short period of time. In the present paper, the radome attenuation is hence corrected by adding 6 dBz to the MRR reflectivity values.

Finally, the equivalent reflectivity, mean Doppler velocity and SNR are hourly-averaged, in accordance with the integration time recommended by Durán-Alarcón et al. (2019) for climatological analysis, in order to remove short time perturbations while keeping enough data for statistical significance. Hours with less than 10 valid minute timesteps were discarded to avoid spurious spikes.

## 2.2 Snow-Gauge

Along with the MRR, an OTT Pluvio$^2$ (model 400cm$^2$) weighing gauge (with a wind shield, see Fig. 2b) was deployed during the austral summer campaign 2015-16 at Dumont d'Urville. Hydrometeors falling into the bucket are measured by a very sensitive weighing system, and converted in mm water equivalent. The weighing gauge hourly snowfall in mm hr$^{-1}$ is then obtained by summing the 1-minute bucket mass changes over one hour. These data are used in Sect. 3.2 to derive MRR snowfall profiles. Various limitations affect the gauge data, which are also discussed in Sect. 3.2. The gauge was deployed in the 2015-16

austral summer campaign, then removed in February 2016 and reinstalled in January 2017 until today. Moreover, it was out of order between December 2021 and December 2022 included. These 2 main gaps are taken into account in the computation of the MRR snowfall estimates in Sect. 3.2. For more information about the DDU snow-gauge, we refer the reader to Grazioli et al. (2017a) and Genthon et al. (2018). The MRR and snow-gauge data range from November 2015 to June 2023, and the instruments are still in operation.



## 2.3 Météo-France Observations

Hourly surface meteorological variables such as 2-m temperature, 10-m wind speed and direction, and 2-m relative humidity with respect to liquid are provided by the Meteo-France weather station at Dumont d'Urville, from January 2015 to June 2023 included. They are used in Sect. 3.2 for the computation of the MRR snowfall profile as a quality-control filter for the weighing gauge data.

## 2.4 Models data

### 2.4.1 The ERA5 Reanalysis

Produced by the European Centre for Medium-Range Weather Forecasts (ECMWF), ERA5 is an atmospheric reanalysis which combines a weather forecast model with meteorological observations from a large number of sources, through a 4-D data assimilation system. It provides various meteorological variables over $\sim 30$ km resolution with 137 vertical levels, from 1979 to nowadays (Hersbach et al. (2020)). ERA5 supersedes ERA Interim, which stopped being produced in 2019.

While surface precipitation has been archived, the forecasted solid precipitation fluxes at the midpoint of the vertical layers in the model have not been saved in the reanalysis archives at the time of the data extraction. To allow comparison with radar vertical profiles of precipitation, we use the method to recalculate ERA5 snowfall rates described in Roussel et al. (2023), Sect. S3. Hourly data from November 2015 to December 2021 included were extracted at the grid point nearest Dumont d'Urville, of coordinates [140.0,-66.75].

### 2.4.2 The LMDZ General Circulation Model

Atmospheric component of the French Institut Pierre-Simon Laplace Climate Model (IPSL-CM) used in particular for the Coupled Model Intercomparison Project exercises, the LMDZ General Circulation Model (Z standing for its zooming capability) is developed at the Laboratoire de Météorologie Dynamique (LMD) in Paris. Hourdin et al. (2020) describes the model's generals, and Madeleine et al. (2020) the clouds and precipitation physics. This model has already been used for various studies in Antarctica (Krinner et al. (2019), Vignon et al. (2018)) and in particular Lemonnier et al. (2021) that opens the way to the evaluation of the representation of the Antarctic precipitation in the model.

We use a simulation ranging from November 2015 to December 2021 included with a 96x95 horizontal grid and 95 vertical levels. The grid is refined around Adélie Land, resulting in a resolution of approximately 50 km in the zoom center (see Fig. S1 of Roussel et al. (2023) for a map of the grid). The run was nudged with wind, temperature and humidity by the ERA5 reanalysis outside of the zoom area. For further details, we refer the reader to the very similar configuration in Roussel et al. (2023) that compared the precipitation at the surface and along the vertical of LMDZ and 5 other climate models with the DDU MRR and snow-gauge during the YOPP (Year Of Polar Prediction) period. We evaluate the model physics used for CMIP-6, which has not been specifically adapted and calibrated on Antarctic precipitation. Hourly precipitation profiles are simply





extracted at the grid cell nearest DDU of coordinates [140.4, -66.63], as the most accurate representation of snowfall amounts is that of the closest grid point regardless of the surface type (Roussel et al. (2023)).

## 3 Statistical analysis

### 3.1 Characteristics of the 7-year record

In this section, before estimating the precipitation flux from the radar data in Sect. 3.2, the hourly-averaged equivalent re-
flectivity Ze, the mean Doppler velocity W and the signal-to-noise ratio SNR are analyzed for the whole 7-year period, from November 2015 to June 2023 included. The derivation of these variables with the MK12 processing method is described in Sect. 2.1.

Fig. 3 presents the daily-averaged 7 years of MRR reflectivity profiles, giving an overview of individual precipitation events and variability with respect to intensity over the whole period. The timeseries is quasi-continuous except for several interrup-
tions identified on the figure by gray zones.

2-D joint distributions of equivalent reflectivity, mean Doppler velocity and SNR are presented in Fig. 4a, 4b and 4c. The percentage of occurrence of those variables along each radar gate is color-coded (left blank for occurrences below 1%), the median is the solid-dotted black line and the 5th, 25th, 75th and 95th quantiles are the gray dashed lines.

The median equivalent reflectivity in Fig. 4a ranges from 5.5 dBz in altitude, then increases as ice crystals grow through
deposition and aggregation and densify through riming (Planat et al. (2021)) towards a maxima of 10.2 dBz at 800 m. Then, reflectivity decreases slightly to 9.7 dBz at 300 m due to snowflake sublimation by low-level dry air blowing from the plateau (katabatic flow). The physics of this process is discussed in Grazioli et al. (2017b). The 95th quantile exceeds 20 dBz in the lower gates, indicating the occurrence of rare but heavy snowfall events. There is no equivalent reflectivity below 1 dBz, as K-band reflectivities lower than -5 dBz have been discarded and an offset of +6 dBz was added to correct for radome attenuation
(see Sect. 2.2). Hence, the radome attenuation correction reduces the MRR sensitivity.

The mean Doppler velocity median decreases with height, going from 1.4 m s$^{-1}$ at 300 m to 0.9 m s$^{-1}$ at 2900 m. Even the 95th quantile does not exceed 2.5 m s$^{-1}$, supporting that rain events are very rare. In fact, only 0.7% of the hourly mean Doppler velocities exceed 3 m s$^{-1}$. Vignon et al. (2021) found that the Dumont d'Urville station experiences in average only
1.8 days of rainfall per year, although that frequency may increase in the next decades.

All quantiles show a sharp increase of approximately 0.2 m s$^{-1}$ at the lowest gate, which is probably due to noise in the signal such as near-field effect, and despite the MK12 data quality masking. The 95th quantile also shows a suspicious increase of the same magnitude at the highest gate (2900 m) probably due to noise in the signal. Most mean Doppler velocities above 2000 m are smaller than 1 m s$^{-1}$. There are a few rare events of negative W in altitude (75 over 12253 hourly timesteps)
corresponding to weak updrafts, whose detection was made possible by the MK12 algorithm (see Sect. 2.1).



**Figure 3.** MRR equivalent reflectivity in dBz. Hatches indicate periods during which the MRR was not in operation. Periods with more than 10% of missing data are in gray shading.

The SNR median in Fig. 4c is constant below 500 m, then decreases steadily with height, going from a ratio of -6.0 dB to -15.6 dB. Most signal-to-noise ratios above 2000 m are below -10 dB. The SNR median is rather low, because the MRR has a high noise level and solid precipitation corresponds to lower reflectivity than rainfall. It should be noted that the SNR 95th quantile ranges between 0 and +10 dB, indicating that heavy events fully stand out from the noise.




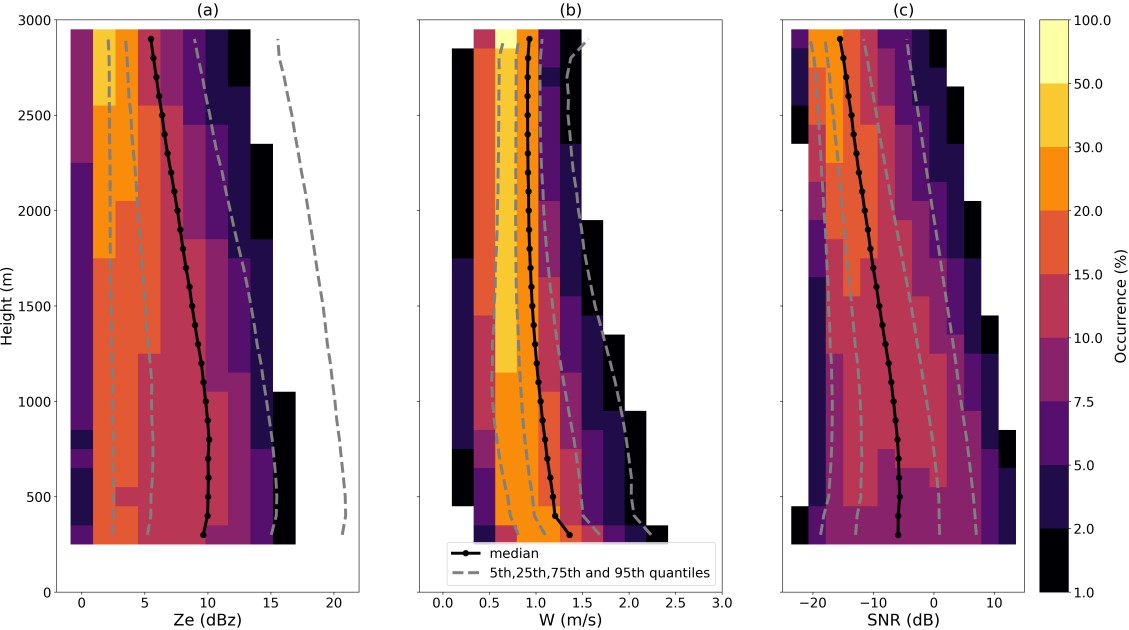

**Figure 4.** Median (black solid dotted line) and 5th, 25th, 75th and 95th quantiles (gray dashed lines) of the equivalent reflectivity Ze (fig. 3a), the mean Doppler velocity W (fig. 3b) and the signal-to-noise ratio SNR (fig. 3c). The shading represents the occurrence along each radar gate (left blank below 1%). Pay attention to the non-linear colorbar.

Overall, the 2D joint distributions shown in Fig. 4 concur with Fig. 4a, 4d and 4g of Durán-Alarcón et al. (2019), which plotted those 3 variables for the same instrument, although for a shorter period (2 years).

The maximum value of the equivalent reflectivity Ze reaches 31 dBz at 1900 m on February 9, 2022 at 6 pm UTC during a
short and intense precipitation outburst of about 10 minutes, surrounded by longer and less intense events. Corresponding mean Doppler velocity exceeds 6 m s$^{-1}$, which suggests the presence of rain in altitude. Unfortunately, this event occurred during local night and rain has not been reported by Météo-France staff at the station, although surface temperatures went above 0°C for a few hours around that time. The snow-gauge was not operating (see Sect. 2.2) and cannot give further information about the magnitude of this event.

A reliable maximum value of the mean Doppler velocity could not be estimated because of various nonphysical peaks detected by the algorithm and resulting from an imperfect dealiasing. Those peaks are not correctly filtered out by the data quality masking of the MK12 algorithm. However, they do not seem to significantly shift the median and quantiles of Fig. 4.b, except for the lowermost and uppermost gates (300 m and 2900 m).

Interannual variability for the 7 years of data is investigated in Fig. 5, which presents the median yearly profiles from 2016 to 2022 of the equivalent reflectivity, mean Doppler velocity and SNR from the hourly-averaged data (8760 data points each

year). Year 2023 is not shown as it was not complete at the time of extraction. We characterize the interannual variability over those 7 years as the maximum profile minus the minimal profile. The equivalent reflectivity variability is 3 dBz (30% of the median profile) and is rather constant along the vertical ; the mean Doppler velocity variability is 0.2 m s$^{-1}$ near 300 m (15% of the median profile) and decreases towards 0.05 m s$^{-1}$ in altitude ; and the SNR variability is 3 dB (50% of the average profile). There is no statistically significant temporal trend of those variables over the 7 years of data (not shown).

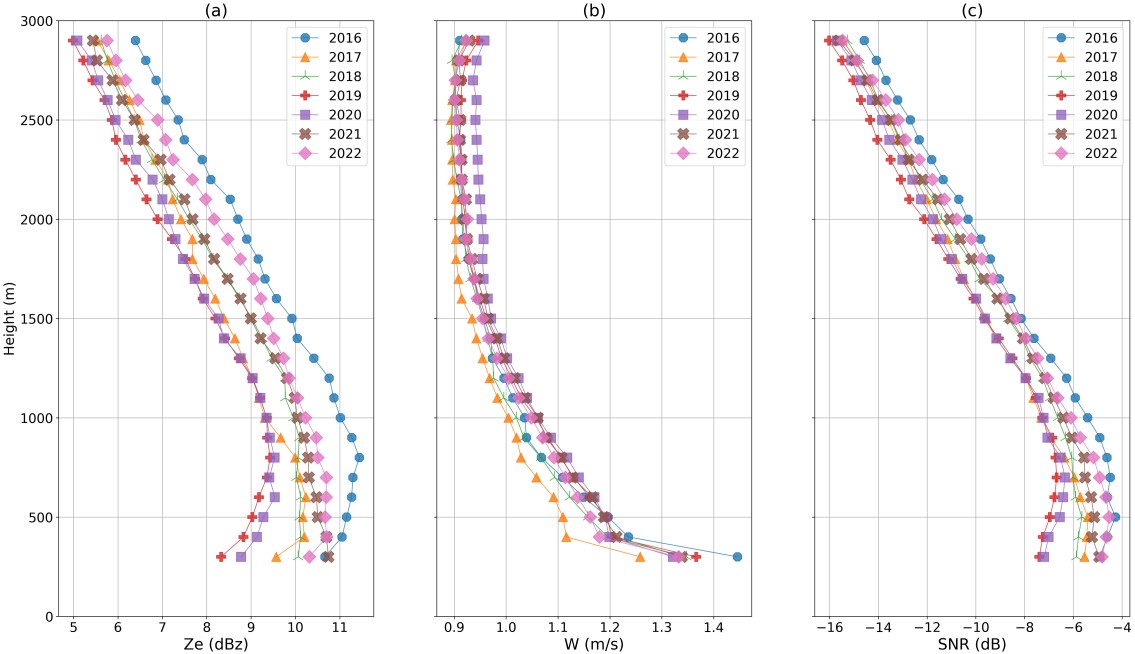

**Figure 5.** Yearly median profiles of the MRR equivalent reflectivity (a), mean Doppler velocity (b) and SNR (c).

Seasonal variability is then considered in Fig. 6 for the same variables, whose medians have been computed for each month over the 7 years of data, thus providing a first estimation of the MRR climatology (around 5500 data points for each month). Here again, no visible trend is identifiable in Figs. 6a and 6c for the equivalent reflectivity and the signal-to-noise ratio. Seasonal variability for those variables, if any, is masked by the substantial interannual variability seen above.

In contrast, a clear seasonal signal of the mean Doppler velocity appears throughout all the column in Fig. 6b. Ice crystals and snowflakes fall slower in winter (June to September) than in summer (December to February), with inter-seasons in the middle values. Indeed, relatively warmer and moister conditions in summer favor aggregation and riming, thus increasing the snowflakes density and fall speed (Garrett and Yuter (2014)). This signal remains when removing one year from the period at a time (not shown), supporting that it is robust to interannual variability.

If we characterize seasonal variability as the maximal profile minus the minimal profile in Fig. 6, the equivalent reflectivity variability is 3.5 dBz (around 40 % of the median profile) at 300 m and decreases slightly with altitude ; the mean Doppler





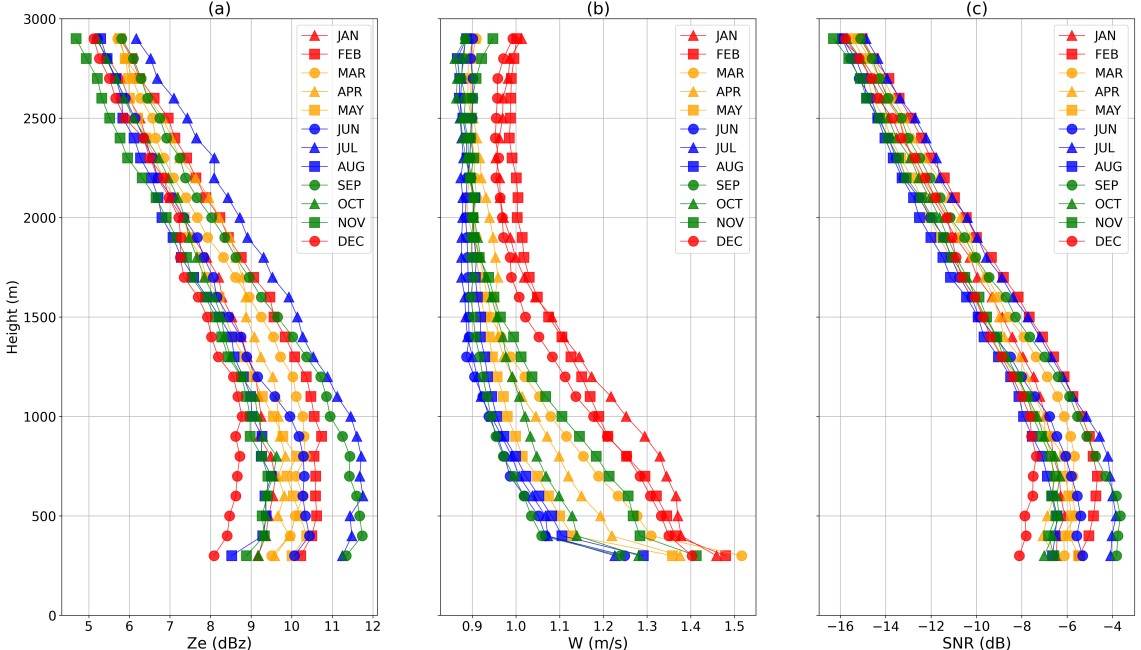

**Figure 6.** Monthly median profiles of the MRR equivalent reflectivity (a), mean Doppler velocity (b) and SNR (c).

velocity variability is 0.3 m s$^{-1}$ at 300 m (21 % of the median profile) and is divided by 2 in altitude ; whereas the SNR
seasonal variability is 4.1 dB at 300 m (70 % of the median profile) and only 1.4 dB in altitude.

Both interannual and seasonal variability are larger near the surface than in altitude. In fact, we expect differences between
precipitation events to emerge during the aggregation and riming processes (Planat et al. (2021)) and thus be more pronounced
in the lower gates. Moreover, turbulence in the katabatic layer (below 1 km or so) is likely to impact the hydrometeors fall
speed, and to increase the mean Doppler velocity variability.

## 3.2 Estimation of snowfall rate profiles from the MRR data

### 3.2.1 Derivation of the Ze-S relation

In this section, the MRR snowfall rate profile is calculated by means of a power law relationship between the equivalent radar
reflectivity Ze (in mm$^6$ m$^{-3}$) and the snowfall rate S (in mm hr$^{-1}$), i.e. $Ze = aS^b$ (eq. 1), whose parameters $a$ (prefactor) and
$b$ (exponent) are estimated using the weighing gauge snowfall S (see Sect. 2.2). This is a common methodology for retrieving
snowfall rates from radar reflectivities (e.g., Grazioli et al. (2017a), Scarchilli et al. (2020), Souverijns et al. (2017)). Theoreti-
cal considerations about snowflakes shape, mass and velocity are discussed in Matrosov (2007) and Matrosov et al. (2009). The
equivalent reflectivity Ze at the usable gate closest to the surface (300 m a.g.l.) is processed as described in Sect. 2.1, except
that it is converted in linear units (mm$^6$ m$^{-3}$) before the hourly averaging. The gauge hourly snowfall is computed as described





in Sect. 2.2. We refer the reader to Sect. 4 for the processing code availability. The regression period spans from November
2015 to June 2023 included, which corresponds to a 5.5 year period when taking into account the two gaps in the snow-gauge
data mentioned in Sect. 2.2. 9456 hourly precipitating timesteps common for both instruments remain to perform the regression.

At this stage, the scatter plot exhibits a large amount of statistical noise with many outliers (see gray dots of Fig. 7). This
issue motivated the application of several filters to reduce the noise. We present and use two of them in the following along with
their impact on the power-law parameters, as they were considered both mandatory and sufficient to obtain a robust regression.

Wind is the main source of uncertainty of snow-gauge measurements at Dumont d'Urville. It makes the gauge vibrate,
destabilizing its weighing system and leading to spurious precipitation records. Moreover, snow having already precipitated
may be remobilized from the surface into the atmosphere by the wind, and fall into the gauge bucket, leading to largely
overestimated snowfall rates and accumulation. Sugiura et al. (2003) showed that blowing snow can lead to an overestimation
of precipitation by 6 to 130% due to the increased number of aeolian snow particles in the atmosphere, a result supported
by numerous very large snow-gauge hourly snowfall rates suspiciously corresponding to low MRR reflectivities. In addition,
blowing snow particles are smaller than snowfall particles (Nishimura and Nemoto (2005), Naaim-Bouvet et al. (2014)) as
they originate from shattered snowflakes, and therefore have a different radar signature since radar reflectivity is dependent on
diameter to the sixth power. This impacts the Ze-S relation as part of the reflectivity signal does not correspond to precipitation.
Hourly wind speed provided by the Météo-France weather station (see Sect. 2.3) is thus used to discard all data points
corresponding to winds above 7 m s$^{-1}$, in accordance with the threshold used in Scarchilli et al. (2020).

As a second filter, weighing gauge hourly snowfall lower than 0.1 mm hr$^{-1}$ and exceeding 12 mm hr$^{-1}$ have been discarded
to avoid respectively biases due to the instrument sensitivity and unrealistic values caused by maintenance operations on the
bucket, the second threshold originating from the snow-gauge documentation.
Data points removed by the application of these two filters are the small gray crosses in Fig. 7. The impact of the filters on
the Ze-S relation is discussed further below.

Hourly surface temperatures provided by Météo-France were also investigated to prevent liquid precipitation (above 0°C)
from impacting the Z-S relation, defined exclusively for snowfall. However, very few data points were affected by this filter
and the impact on the parameters was negligible, which is consistent with the rare rainfall occurrence found in Sect. 3.1 from
the mean Doppler velocity. Therefore, this filter was not retained in the final Ze-S relation computation.

Daily occurrences of blizzard and blowing snow from Météo-France weather reports were also tested as a filter, but proved
too coarse to efficiently clean out the noise in the correlation cloud. Again, this filter was not retained in the computation.

Only 503 data points are left after the application of these two filters, but they are quite evenly spread over the whole period
(not shown). A significant amount of information is thus lost in the filtering process, but we consider it mandatory to extract
the signal from the noise.

The 503 data point cloud resulting from the two filters described above still visually exhibits a dozen outliers, likely to affect
fits based on root mean square error minimization. It was therefore decided to convert equation (1) in log space, and to fit the

resulting equation :

$$ln(Ze) = b * ln(S) + ln(a) \tag{2}$$

with a quantile linear regression method more robust to ouliers compared to the standard least-square method (Pedregosa et al. (2011)).

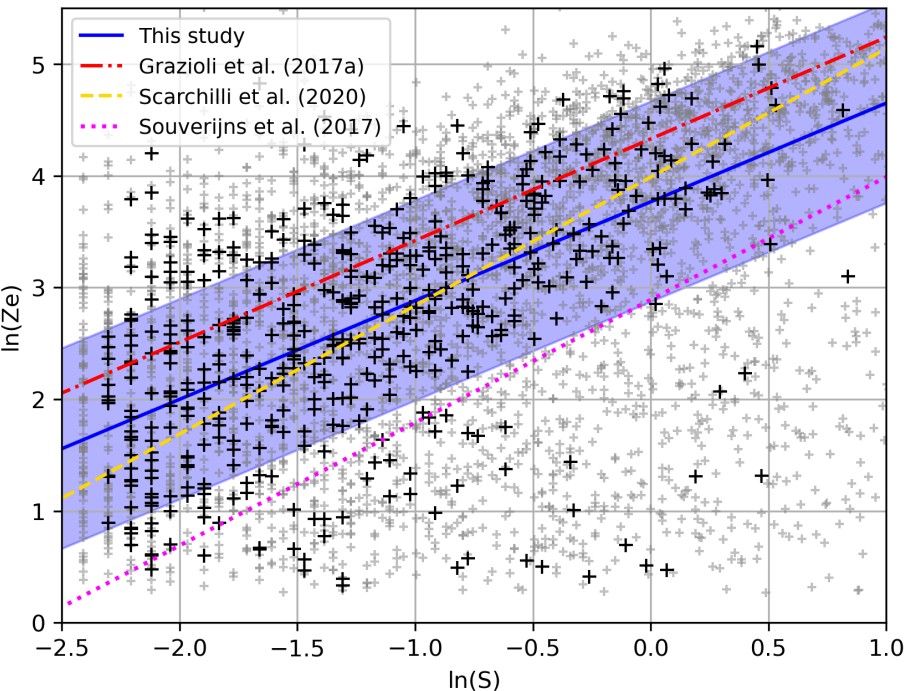

**Figure 7.** MRR equivalent reflectivity and gauge snowfall rate scatter plot (black crosses) in log space filtered as described above, and the resulting Ze-S relation derived with a robust quantile regression (blue solid line). The blue shading represents the regression RMSE. Ze-S relations from Grazioli et al. (2017a) (red dashed line), Scarchilli et al. (2020) (yellow dashdot line) and Souverijns et al. (2017) (dotted magenta line) are also represented. Small gray dots represent data points that have been filtered out.

Figure 7 presents the resulting scatter plot after filtering and the linear fit in log space. Converted back into linear space, the relation obtained is Ze = 43.3 $S^{0.88}$, with a $R^2$ score in log space of 0.27. We estimate the regression uncertainty as the RMSE,

plotted in blue shading on Fig. 7. When the filter based on the wind speed threshold is deactivated, there is much more noise in the scatter plot and the $R^2$ score drops to 0.10, whereas parameters $a$ and $b$ change by -5% and -20%. Thus, this filter allows to remove a significant amount of outliers corresponding to high snow-gauge snowfall and low MRR reflectivity values typical of blowing snow conditions, with a major impact on the $R^2$ score and the regression slope $b$ (see equation (2)). When the filter based on the snow-gauge values is deactivated, parameters $a$ and $b$ decrease by -26% and -32%. Although the $R^2$ score is better





(0.33) without this filter, it was retained as the weighing gauge snowfall very low values with high relative incertitude (>10%) lead to a less realistic regression, as can be seen by the significant impact on the parameters. We made the choice of converting the Ze-S relation in log space and using a robust quantile linear regression, as by reducing the impact of outliers it leads to a regression that better fits the data, with almost no change to the $R^2$ score (<0.01). Parameter $a$ ($b$) changes by +17% (+15%) compared with the standard linear regression, which is quite significant.


    Compared to the Ze-S relation Ze = 76 $S^{0.91}$ found by Grazioli et al. (2017a) (plotted in dashed red on Fig. 7) from the same instruments although for a much shorter period, the slope $b$ is very similar (inside the 95th confidence interval [0.78-1.09]) while the fit has an offset due to the much lower prefactor (outside the 95th confidence interval [69-83]), leading to a negative $R^2$ score. When our Ze-S relation is derived over the same period as Grazioli (from 2015-11-22 to 2016-01-29), although the

small number of data points does not allow to apply the filters, the two parameters fall back inside the 95% confidence interval : $a$ = 73.7 and $b$ = 1.05. Small discrepancies persist due to different data processing methods such as the conversion of the Ze-S relation in log space, the use of a robust quantile linear regression, or the initial processing of S and Ze. Over the whole period, the parameters found in this study lead to higher MRR snowfall values than Grazioli et al. (2017a), as a given equivalent reflectivity corresponds to a higher gauge snowfall. These considerations could be of interest for future studies using the DDU

MRR as an observation reference, as up to now, only parameters from Grazioli et al. (2017a) derived for a two-month period in austral summer have been used (e.g., Lemonnier et al. (2019), Jullien et al. (2020), Roussel et al. (2023)).

    The Ze-S relation of Scarchilli et al. (2020) Ze = 54 $S^{1.15}$ (although for an integration time of 5 minutes) in dashdot yellow on Fig. 7 fairly well fits the data, although its $R^2$ score is lower (0.20) as it does not take into account the outliers located in the lower right corner of the scatter plot. Souverijns et al. (2017) relation Ze = 18 $S^{1.10}$ in dotted magenta is outside our

uncertainty range in blue shading, with a negative $R^2$ score. Likewise, our parameters $a$ and $b$ are both outside their uncertainty range ([11-43], [0.97-1.17]). This significant difference can be due to the location of the Princess Elizabeth station, 173 km off the coast at the other end of Antarctica with drier conditions and smaller particle diameters ; the much shorter sampling period (January to May 2016), or the instrument they used as snowfall reference (the Precipitation Imager Package). The prefactor value $a$ = 43.3 found by this study is consistent with those of the theoritical Ze-S relations of Matrosov et al. (2009) which

range from 28 to 136, although the corresponding exponents exceed 1. However, definitive conclusions cannot be drawn for $a$ or $b$ individually, as they are not independent.

    Despite the restrictive filtering steps listed above, the regression score $R^2$ in log space remains low (0.27), which is mainly due to outliers in the lower right corner of the scatter plot and probably not linked with blowing snow as they have not

been filtered out by the wind speed threshold. Furthermore, the precipitation flux at DDU may be explained by other factors than radar reflectivity. Other explanatory variables, potentially season-dependent, could be used to derive better precipitation estimates. This important issue will be addressed again in Sect. 4.



### 3.2.2 Sensitivity to integration time

Interestingly, the Ze-S relation obtained with the data processing detailed previously is robust to the integration time choice.
Table 1 presents the values of parameters $a$, $b$, the number of data points N and the score $R^2$ for different integration times. The hourly wind speed filter from Météo-France was linearly interpolated in time when the integration time was below 60 minutes. Although the shorter (5 minutes) and longer (180 minutes) integration times differ from the average with lower $R^2$ scores due to respectively too much noise and undersampling effects, the parameters $a$ and $b$ are approximately independent from the integration time. The hourly integration time exhibits the higher $R^2$ score.


| Integration time (min) | a | b | N | $R^2$ |
|---|---|---|---|---|
| 5 | 48.5 | 0.87 | 5712 | 0.19 |
| 15 | 44.6 | 0.88 | 1756 | 0.20 |
| 30 | 45.0 | 0.94 | 939 | 0.25 |
| 60 | 43.3 | 0.88 | 503 | 0.27 |
| 90 | 43.0 | 0.85 | 335 | 0.23 |
| 120 | 44.0 | 0.81 | 258 | 0.22 |
| 180 | 38.5 | 0.68 | 179 | 0.20 |

**Table 1.** Impact of the integration time on parameters $a$ and $b$, the number of data points N and the $R^2$ score.

| Year removed | a | b | N | $R^2$ |
|---|---|---|---|---|
| 2016 | 43.1 | 0.88 | 498 | 0.27 |
| 2017 | 43.2 | 0.95 | 418 | 0.30 |
| 2018 | 42.3 | 0.89 | 405 | 0.23 |
| 2019 | 43.2 | 0.86 | 405 | 0.23 |
| 2020 | 43.6 | 0.85 | 377 | 0.24 |
| 2021 | 43.2 | 0.89 | 458 | 0.33 |
| 2022 | not in operation | | | |
| 2023 | 44.1 | 0.86 | 464 | 0.27 |

**Table 2.** Impact of the interannual variability on parameters $a$ and $b$, the number of data points N and the $R^2$ score.

Sensitivity of the Ze-S relation to interannual variability was also assessed by removing one year at a time in the computation. The prefactor $a$ varies between 42.3 and 44.1, whereas the exponent b varies between 0.85 and 0.95 (see Table 2). As these variations remain within 10% relative difference, this result supports that the derived Ze-S relation is also robust to interannual changes in the recordings.



In Sect. 3.3, the MRR snowfall profile is computed by inverting the equation $Ze = aS^b$ :

$$S = (\frac{Ze}{a})^{1/b} \tag{3}$$

with the parameters found for the whole dataset with an integration time of 1 hour. Doing so, we assume that the Z-S relationship derived using the MRR data at 300 m is representative for the entire profile (up to 3 km), which may not hold true when there is a large change in the hydrometeors structure and type along the vertical.

## 3.3    Example of application to models evaluation

     In this section, we show how the hourly MRR snowfall computed from the Ze-S relation described in Sect. 3.2 can be used to evaluate the vertical profiles of precipitation as simulated by numerical models. The MRR is compared to the ERA5 reanalysis and the LMDZ model (described in Sect. 2.4) for the period ranging from 2015-12-01 to 2022-01-01.

     Firstly, the median and quantiles profiles of the three datasets in coastal Adélie Land are investigated. The model profiles

are precipitation threshold-sensitive, as they produce a large number of very small precipitation events (below 0.01 mm hr$^{-1}$). Palerme et al. (2014) proposed a threshold of 0.07 mm per 6 hours to optimize the comparison of ERA-Interim precipitation rates with CloudSat observations, which was also used in Roussel et al. (2023) for models evaluation. Converted to mm hr$^{-1}$, the models' threshold is 0.012 mm hr$^{-1}$. The MRR snowfall threshold is derived from the equivalent reflectivity sensitivity of -5 dBz, i.e. +1 dBz taking into account radome attenuation (see Sect. 2.1) and $10^{1/10}$ = 1.3 mm$^6$ m$^{-3}$ in linear units. Equation

(3) with the parameters a and b found in Sect. 3.2 gives a MRR threshold of 0.019 mm hr$^{-1}$. To avoid sampling period biases due to MRR missing data, only precipitating times (i.e., with a precipitation rate above the thresholds defined above) for the three datasets have been retained, corresponding to 3717 data points. Figure 8 shows the profiles medians (solid lines with dots) and the 10th and 90th quantiles (dotted lines).

     The MRR 90th quantile stands out rather strikingly, peaking more than twice as large as the corresponding ERA5 and LMDZ

profiles, while its median and 10th quantile remain of similar magnitude. This result suggests that models struggle to simulate heavy events. Maximum precipitation values reached at the surface (at 300 m for the MRR) for that period are consistent with that statement : ERA5 attains only 2.7 mm hr$^{-1}$ and LMDZ 4.6 mm hr$^{-1}$, whereas the MRR reaches 30 mm hr$^{-1}$.

     Although ERA5 and LMDZ reproduce the increase in precipitation from the top (3 km) down to a maximum around 800 m, as well as the decrease below due to sublimation of snowflakes, the slopes are not steep enough compared to the MRR.

This suggests that auto-conversion of ice crystals into snowfall as well as sublimation due to the katabatic flow may be underestimated in the models, an issue already raised for ECMWF-IFS and LMDZ by Grazioli et al. (2017b) with one year of data of the same instrument. Yet, the altitudes of maximum precipitation in ERA5 and LMDZ fairly well correspond to the observations, with an ERA5 median profile peaking at 890 m and LMDZ at 740 m, while the MRR peaks at 800 m. Although the MRR data do not extend below 300 m, the slopes of the model profiles below 800 m substantially overestimate surface

precipitation because of too weak sublimation. If sublimation is defined as the relative difference of the median snowfall rate between the altitude of maximum precipitation and 300 m (including for the models), the MRR sublimation is 40%, whereas



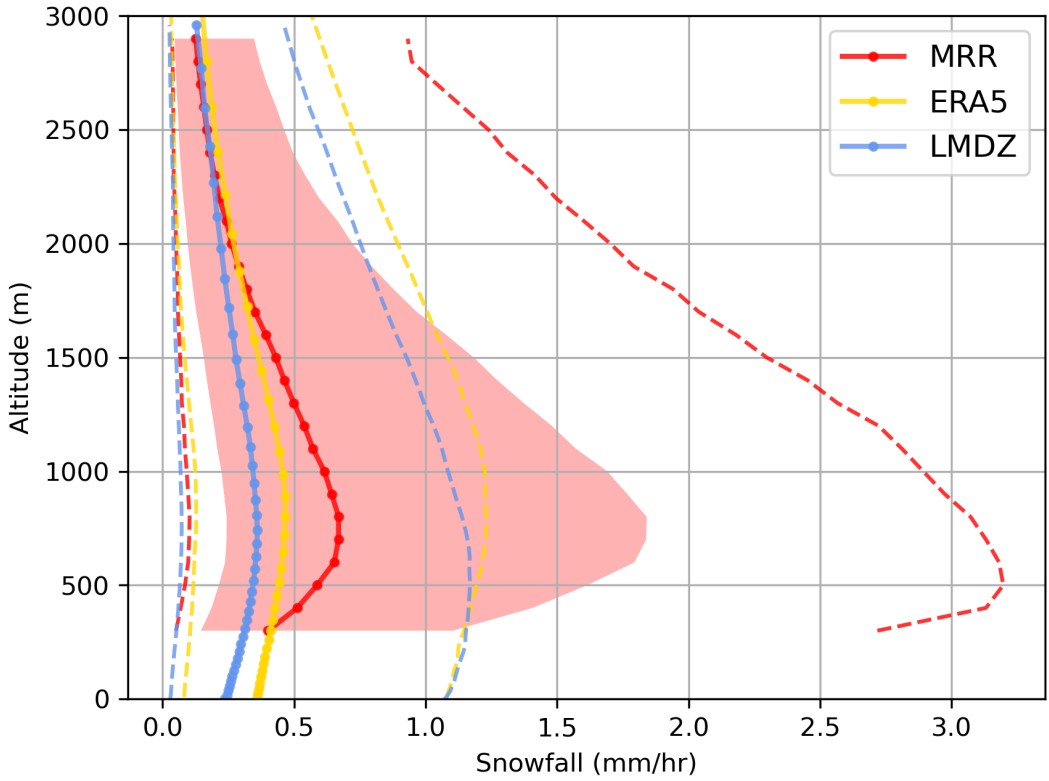

**Figure 8.** Median (solid line with dots), 10th quantile and 90th quantile (dashed line) vertical profiles for the MRR (in red), ERA5 (in yellow) and LMDZ (in blue) for the 3717 hourly data points of common precipitating timesteps.

ERA5 (LMDZ) sublimation is only 12% (13%).

The underestimation of strong events intensity by the models has a big impact on total snow accumulation. Indeed, Fig. 1
of Turner et al. (2019) indicates that extreme precipitation events contributes by more than 50% to total accumulation near the
DDU region. In the present dataset, almost 50% of the MRR total accumulation is due to snowfall larger than 2 mm hr$^{-1}$,
whereas this contribution drops below 5% for ERA5 and LMDZ (not shown). This result is consistent with Fig. 11 of Grazioli
et al. (2017a) (upper panel) for one year of the same instrument, although the contribution of MRR snowfall rates larger than
2 mm hr$^{-1}$ is only 30%. This difference can be explained by the larger MRR snowfall rates found by this study through the
revised Ze-S relation (see Sect. 3.2).

At the end of the period, after having removed the gaps in the MRR dataset, the MRR yearly average accumulation at 300m
is 1060 mm yr$^{-1}$. It is larger than ERA5 by 44% with 737 mm yr$^{-1}$ and than LMDZ by 53% with 691 mm yr$^{-1}$ for the
same period and altitude. However, the models' accumulation at 300 m may be underestimated due to their too smooth profile
shape (see Fig. 8). Conversely, the MRR snowfall rate may be overestimated due to remaining outliers in the Ze-S computation





despite the restrictive processing method described in Sect. 3.2. Grazioli et al. (2017a) estimated yearly accumulation from October 2015 to October 2016 between 740 and 989 mm yr$^{-1}$, i.e. 23% less in average than this study, which is consistent again with lower snowfall rates obtained with the Grazioli et al. (2017a) Ze-S relation. Even so, precipitation has a high degree of interannual variability (see Fig. 5a) and two different periods cannot be directly compared. Also, it is important to keep in mind that the accumulation at the surface is probably much lower than 1060 mm yr$^{-1}$ because of precipitation sublimation

below the MRR lowest gate at 300 m. Snow-gauge accumulation is not presented here as it is largely overestimated due to contamination by blowing snow (not shown).

On the other hand, models precipitation occurrence is higher than the MRR, with 29% of hourly precipitating timesteps for ERA5, 22% for LMDZ and only 15% for the MRR. Model snowfall events are also longer with a median of 13h (14h)

for ERA5 (LMDZ) compared to 7h for the MRR. However, this higher occurrence of model precipitation is not enough to compensate for the larger MRR accumulation.

## 4 Conclusions

7 years of data from a Micro Rain Radar (MRR) deployed at the Dumont d'Urville station in Antarctica are presented. A statistical analysis outlines the main characteristics of the MRR vertical profiles of the equivalent reflectivity, mean Doppler

velocity and signal-to-noise ratio, concurring with the results of Durán-Alarcón et al. (2019). No interannual or seasonal trend have been clearly identified in the MRR profiles, except for the seasonal mean Doppler velocity which is larger in summer and smaller in winter, suggesting an enhanced aggregation process. Nonetheless, the sample period is still short (7 years) to possibly exhibit such climatic trends.

A Ze-S relation has been derived from the dataset to retrieve precipitation profiles, thus allowing to refine the relation found

by Grazioli et al. (2017a) for the same instrument but built on one summer season only. Despite a large amount of noise, the 7-year period made it possible to apply restrictive filters robust to integration time and interannual variability. The uncertainty of the Ze-S relation is estimated as its RMSE in log space. The results have been compared with literature, and particularly with the relation of Grazioli et al. (2017a) with whom an offset probably due to the sampling period has been found, leading to smaller MRR snowfall rates than this study. However, Ze-S relations in the literature still present a significant degree of

uncertainty and makes it difficult to draw final conclusions. Although we chose to be very cautious, the R$^2$ score remains low and other processing methods may be applied to the raw data (see data availability below) by future users. For instance, other explanatory variables such as temperature, wind speed, or the mean Doppler velocity of hydrometeors could be combined to the equivalent reflectivity to better constrain the Ze-S relation, but such considerations are beyond the scope of this study.

This Ze-S relation allowed the evaluation of two climate models (ERA5 and LMDZ) along the vertical as an application

example of the dataset. Models showed profiles too smooth, both in altitude and time, with a large underestimation of intense snowfall events compared to the MRR leading to an accumulation twice as small. The weaker sublimation of precipitation by the models is not enough to compensate for their smaller accumulation.

We believe that the 7 years of data presented in this paper are a great opportunity to evaluate and optimize climate models by fostering future studies on the parameterization of snowfall along the vertical in Antarctica, as well as the representation of the
katabatic layer and its impact on precipitation sublimation. The dataset can also be used to complement and validate satellite products by providing ground-based information, for instance to evaluate the effect of the blind range near the ground level to obtain more accurate surface precipitation estimates.

*Data availability.* Data are available here : https://web.lmd.jussieu.fr/~vwiener/MRRDATA/ (Wiener et al. (2023)), and are furthermore currently under review on PANGAEA. A DOI and full data citation will be provided at the end of the process. The MRR 1-min profiles of
the source variables (equivalent reflectivity, mean Doppler velocity, SNR and quality flag) are stored in zipped netCDF files from 2015-11-23 to 2023-07-01 (one file per year). The MRR hourly snowfall profiles computed in Sect. 3.2 from the relation Ze = 43.3 S$^{0.88}$ along with the gauge 1-min snowfall accumulation and the hourly wind speed and temperature from Météo-France observations are also available for the same period. The processing code in python used for the derivation of the Ze-S relation is also attached as a Jupyter Notebook.

*Author contributions.* **Valentin Wiener:** resources, data curation, investigation, methodology, writing. **Marie-Laure Roussel:** resources,
review. **Christophe Genthon:** instrument setting, writing, review, validation. **Étienne Vignon:** investigation, review. **Jacopo Grazioli:** instrument setting, methodology, review. **Alexis Berne:** instrument setting, methodology, review, validation.

*Competing interests.* The authors declare no conflict of interest.

*Acknowledgements.* The LMDZ simulation was performed using HPC resources from the IDRIS (Institut du Développement et des Ressources en Informatique Scientifique, CNRS, France), projects RLMD AD010107632R1. This study benefited from the ESPRI computing and data
center (https://mesocentre.ipsl.fr), which is supported by CNRS, Sorbonne Université, Ecole Polytechnique, and CNES as well as through national and international grants. We thank Claudio Durán-Alarcón for sharing his script for the MRR raw data processing, and Jean-Louis Dufresne for sharing the DDU snow-gauge data, with an instructive critical insight. We also thank IPEV (the French polar institute) for support to program CALVA, and CNES for support to program EECLAT. Finally, we gratefully thank Pauline Jaunet, Laurent Baudchon and Météo-France for preparing and sharing the DDU weather station data.



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
