# Peer review of "A 7-year record of vertical profiles of radar measurements and precipitation estimates at Dumont d'Urville, Adélie Land, East Antarctica"

_Earth System Science Data, 2023_

## Author Response (AR1)

**Reply to Referee Comments**

20/10/2023

(review in black, authors response in red)

Firstly, the authors would like to thank the reviewers for having read and provided constructive feedback on the manuscript.

**RC1**

However, some minor aspects could be reviewed and clarified before publication:

Radome attenuation: The MRR at DDU is placed in a radome. Radome attenuation was estimated at 6 dBZ, which was added to MRR observations to correct them. This is a key point. Have you taken into account the attenuation of a snow layer that can form over the radome during heavy precipitation? Or does the strong wind help to keep the radome clear? Another point: is the radome heated? If yes, the possible snow melting over the dome during precipitation strongly increases the attenuation of the radar signal. Have you made some tests in this respect?

Thank you for raising this important issue. The radome has a domed shape (as you can see on the attached picture radome_MRR.JPG) that prevents the snow from accumulating, especially with the frequent strong winds conditions. When asked, the winter-over staff reported never having seen any snow accumulation so far. As the radome is not heated (except for a small heated cabinet inside which protects sensitive parts of the radar electronics), snow melt cannot occur either.

We therefore mentioned that important point in the manuscript.

Sublimation: In lines 30 and on, you dealt with the sublimation processes during precipitation, a well-known feature of the snowfall over the Antarctic coast. You have cited Grazioli (2017), in which an average snowfall reduction of 17% was estimated at DDU. More recent works (Alexandre,2023 10.1029/2022JD038389 and Bracci,2022 https://doi.org/10.23919/AT-AP-RASC54737.2022.9814266) have found higher values of snowfall reduction (up to 50%) at Davis station and Mario Zucchelli station, respectively.

Thank you for the information, we added those references to the text. However, the 17% obtained by Grazioli et al (2017) were computed over the whole Antarctic continent, not at DDU only ; it is therefore not surprising that this averaged value is much lower than Alexander et al (2023) and Bracci et al (2022). Grazioli et al (2017) furthermore mention that sublimation is up to 35% on East Antarctica margins, which is much closer to the references cited above.

Have you used your extended dataset to calculate a more robust estimation of snowfall reduction at DDU compared to Grazioli? It would be interesting to do this as the decrease in snowfall rate due to sublimation has a fundamental impact on comparing snowfall estimation achieved from ground-based instruments, satellites, and numerical models.

Yes, we mentioned in Sect. 3.3 (line 375 of the revised manuscript) that the MRR sublimation defined as (MRR[zmax] – MRR[300m]) / MRR[zmax] is about 40%, which is quite similar to Alexander et al (2023) and Bracci et al (2022), whereas the models used in our study showed sublimation only slightly above 10%. However, we chose not to dig deeper into that question as this process is not the main focus of the datapaper.

Rainfall episode: In lines 194 and on, some hints of the presence of rain in altitude are reported. It is a very interesting event. Have you sought confirmation (satellite, numerical models…)?

There are very few data sources that can give information about condensed water in the atmosphere of Antarctica. The Cloudsat satellite, even if passing by exactly at the right moment, could not have given information as it operates daytime only since 2011, and as 6 p.m. UTC at DDU corresponds to nighttime. As for the models, ERA5 missed the event and the LMDZ run did not extend to 2022 (but we do not trust much its temporal accuracy anyway as the model agrees only 40% of the hourly timesteps with the MRR on whether there is precipitation or not (not shown in the paper)). Moreover, as mentioned in the paper, the pluviometer was unfortunately out of operation in 2022. Therefore, there is to our knowledge no other way to confirm or invalidate the DDU MRR observations that particular day.

Spectrum width: lines 226 and on, even the spectrum width (recorded by MRR) could be used to highlight the presence of turbulence in the lower levels (That's more of a suggestion).

Absolutely, it has in fact been done in Vignon et al (2020) (see Appendix A) to quantify the turbulent kinetic energy dissipation. Added to the text.

Hourly timesteps: For Ze-SR calculation, if I have understood correctly, you have at first 9456 hourly precipitating timesteps that, after applying filters for wind speed and weighing gauge issues, decrease to 503. It is a very consistent reduction of data. So you extract the "mean" Ze-SR relationship for DDU site, but, at the same time, the signal representing the intrinsic variability of snow is lost. I would suggest you also calculate Ze-SR relationships by not applying filters or using a smoother filter for windspeed to get a sense of the variability of Ze-SR relationships in such an extended dataset for an Antarctic site.

The Ze-S relation without any filter is Ze = 38.8S^0.63, which is substantially different from the relation with the filters. We chose to apply these restrictive filters because although we agree that the reduction of data is quite significant, we did not trust a Ze-S relation over a cloud point so scattered (see gray crosses in Fig. 7). Using a wind threshold for deriving Ze-S relations has already been used in Schoger et al (2021), Scarchilli et al (2020) and others to prevent snow gauge data from overestimating precipitation by undercatching blowing snow and be sure that we were looking at pure snowfall events.

References: In the Introduction, I would suggest reviewing this section by adding some references because some statements need them (e.g., line 15 or 65).

Added some references to statements in the introduction (Christopher et al (1997), Krinner et al (2007), Alexander et al (2023), Bracci et al (2022), Church et al (2008), Di Natale (2022), Peters et al (2002), Maahn and Kollias (2012)).

In the list of MRR installations over Antarctica, I would include the MRR at Concordia (Di Natale, 2022, https://doi.org/10.5194/amt-15-7235-2022) and at Davis station (Alexandre,2023 10.1029/2022JD038389 ) station as they are new deployments over the Antarctic continent, making the Antarctic MRR network grow.

Thanks for the information, added to the text.

Coordinates: Harmonize lon/lat coordinates along the text. In some cases, the lon/lat is made explicit in other no. Maybe it could be explained only at first.

Ok, modified.

**RC2**

**Specific comments:**

- The study is well cited with previous literature. It would be nice to see more references outside of the Antarctic region and what appears to be a tight community with circular references. Although, I also recognize that the applicability of most other studies using K-band precipitation observations to be of limited use in the Antarctic.

Indeed, as the MRR was originally designed to measure liquid precipitation, the MRR community about solid precipitation is much smaller.
Yet, we included in the manuscript Schoger et al (2021) that computed a Ze-S relation in the K-band from a MRR deployed in Ny-Ålesund, Svalbard, Norway. Interestingly, the relation better fits our data than other stations in Antarctica (see fig7_with_schoger.png), possibly due to more similar temperature conditions.
Some other non-Antarctic references were added in the text, such as Chellini et al (2022) who applied the Maahn and Kollias (2012) processing method for a MRR also deployed in Ny-Ålesund.

- The opening line of the abstract is a little strong, especially the second clause in regards to mitigating sea-level rise, than what is necessary for a manuscript regarding observations of precipitation by a MRR at Dumont d'Urville. While I agree that precipitation measurements are significant to Antarctic ice sheet mass balance I would view it as more a point of background/motivation than a summary point in the abstract of this manuscript.

Noted, changed to :

"Studying precipitation falling over Antarctica is crucial as snowfall represents the main water input term for the polar cap. However, precipitation observations still remain scarce, and more particularly in the atmospheric column, due to various experimental issues specific to the white continent.

This paper aims at helping to close this observation gap by presenting 7 years of Micro Rain Radar (Metek MRR-2) data..."

- The figures for Sec. 3.1 provide a good overview and representation of the MRR dataset. I think they are limited in their usefulness for any specific science content but that is fine given that they provide a great summary of the collected data.

Noted, thanks for the feedback.

- I found the inclusion of calculations to identify significant trends in the MRR observations over a period of seven years to be unnecessary. I guess that is a general component in the current world of trying to identify climate impact trends whenever possible, even in relatively short duration datasets.

While we agree that climatic trends cannot be easily derived from a 7-years period dataset – and especially for precipitation measurements, seasonal trends could be highlighted with some robustness as is the case with the mean Doppler velocity (see Sect. 3.1).
In fact, we present our dataset's trends more as statistical characteristics than potential signals related to global warming.

- I found it interesting that about one-fourth of the text in the manuscript was devoted to Sec. 3.2 and the Ze-S relationship. Yet the determination of prefactor and exponent parameters is one of the most significant aspects of the quality and usefulness of the data. The one lingering question that I had is just how much does the variation in the parameters impact the results. For example, if Grazioli et al. (2017a) is used instead of the values from this manuscript, by how much does that change the results in Fig. 8.

Indeed, the parameters' values have a significant impact on the resulting snowfall. If the Grazioli et al (2017) relation is used to obtain Fig. 8 (see fig8_with_grazioli.png), the MRR precipitation profiles and yearly accumulation (not shown) are almost halved. This is why we were particularly cautious in the processing and filtering of the Ze-S relation, which was anyhow expected to be different from Grazioli et al (2017) given the very different operation periods.

However, our paper's conclusions regarding the models underestimation of heavy snowfall events, the duration of events and sublimation in the lower layers are not affected.

**Technical corrections:**

L38:    Include "power" as in "low power consumption" to clarify what is being referenced with consumption.

L86:    Include a reference for the YOPP-SH activities.

L142-144: The opening sentence in this paragraph is difficult to follow. At a minimum, it should be split into two sentences. I am not entirely certain what is trying to be said in the opening sentence.

L153: Do not need to spell out YOPP given that it was previously defined on L86.

L153: Define CMIP6 and include a reference either for the general CMIP6 studies or for any specific MIP that would be of relevance for Antarctic precipitation.

L180: Either include a reference to the statement of the frequency increasing in the next decades or remove the last clause of this sentence.

Authors agree with all technical corrections and have taken them into account in the manuscript.